# Ultrasound, Histomorphologic, and Immunohistochemical Analysis of a Cardiac Tumor with Increased Purkinje Cells Detected in a Canine Fetus 42 Days into Pregnancy

**DOI:** 10.3390/vetsci11050216

**Published:** 2024-05-13

**Authors:** Enrico Giordano, Ignazio Ponticelli, Simona Attard, Teresa Bruna Pagano, Maria Carmela Pisu

**Affiliations:** 1Veterinary Clinic Giordano-Ponticelli, 80011 Acerra, Italy; ambvetacerra@libero.it (E.G.); iponticelli@fastwebnet.it (I.P.); 2VRC—Veterinary Reference Center, 10138 Turin, Italy; mariacarmela.pisu@vierreci.it; 3MyLav Laboratory, Via Giuseppe Sirtori, 9, 20017 Passirana di Rho, Italy; teresabrunapagano@mylav.net

**Keywords:** heart disease, puppy heart tumor, Purkinje cell tumor, immunohistochemistry

## Abstract

**Simple Summary:**

This study examined a case of cardiac tumor with increased Purkinje cells found in the heart of a neonate dog, a condition primarily observed in humans and swine. During a pregnancy ultrasound involving a Chow Chow, a mass was detected in one fetus’s heart, resulting in the birth of a stillborn neonate. Post-mortem examinations revealed the poorly demarcated mass. Histology and immunohistochemistry revealed a complex mass with Purkinje cells that were increased in number and concentrated in perivascular areas. This study emphasizes the uniqueness of diagnosing a cardiac tumor in a dog fetus during pregnancy. Furthermore, the description of a Purkinje cell-rich cardiac tumor is extremely rare in the veterinary literature. The findings of this study draw parallels between Purkinje cell tumors in dogs and those observed in humans, swine, bearded seals, bovine, and deer.

**Abstract:**

A seven-year-old healthy female Chow Chow was referred for pregnancy monitoring. Ultrasonography was used to evaluate all pregnancy and fetus parameters, and they were found to be normal. During the examination of the 42 day pregnant bitch, an unusual mass was seen in a fetus’s heart. This fetus had a cardiac frequency of 273–300 beats, while the others had heart rates of 220–240 beats. Natural vaginal birth occurred at 63 days pregnant: the first two puppies were stillborn but perfectly formed, and the other three were alive and had optimal APGAR. In one of two deceased puppies, an unusual, reddish, smooth mass occupying the space in the heart was found through necroscopy. The organ was submitted for histological examination. Histopathology, immunohistochemical, and histochemical analyses all indicated a cardiac tumor with increased Purkinje cells. This type of tumor has been described in infants, swine, bearded seals, and deer but never in fetuses and neonates of dogs. To our knowledge, this is the first such case reported in veterinary medicine.

## 1. Introduction

Cardiac tumors are uncommon neoplasms in humans. Infants have a higher incidence of primary cardiac neoplasms compared to adults [1,2]. The veterinary literature indicates that this tumor is less common in senior dogs than in young ones [3]. Rhabdomyoma is a type of cardiac tumor observed in young humans or animals. One subtype of the large group of cardiac tumors is the Purkinje cell tumor. In human medicine, it is classified as histiocytoid cardiomyopathy or Purkinje cell hamartoma [1]. Sakurai and colleagues described a multifocal increase in Purkinje fibers in a 4 month old calf, [4]. a condition sharing many histological similarities with the present case; however, in contrast to our report, it was a diffuse lesion without a true mass disrupting myocardial architecture. The tumor is found in infants, swine, and bearded seals but never in other species, including neonate dogs [5]. In human medicine, infants affected by this tumor become unresponsive and can die suddenly [6]. This study aimed to report the ultrasound characteristics and pathological findings of a cardiac tumor with increased Purkinje cells in the heart of a neonate dog. 

## 2. Case Description

### 2.1. Clinical Findings

A seven-year-old healthy female Chow Chow dog was referred to reproduction specialists to monitor her pregnancy. The exam was conducted using a GE Logiq V2 ultrasound machine (GE Health Cre, Milan, Italy)equipped with a 12 LRS linear probe and an 8 CRS micro-convex probe. The ultrasonographic exam (US) was performed on day 25 of pregnancy, and all parameters were within physiological limits. Progesterone concentration was measured using the Tosoh AIA360 (Tokyo, Japan) analyzer due to a history of hypoluteinism. The results were 34 ng/mL at diagnosis and 36 ng/mL one week later. On day 42 of pregnancy, the US detected the following parameters: anechoic liquid in all neonates without echoic smoke and all fetuses showing growth rates consistent with the pregnancy date. Only one of the five fetuses presented an abnormal, echoic mass inside its heart (Figure 1). It was located in the left atrioventricular junction, displaying a soft fringe and a light shadow cone. The diameter measured was 0.49 cm on the puppy’s dorso-ventral axis and 0.47 on the craniocaudal axis. The fetus’s heart rate (HR) was 273 beats per minute (BPM). All other organs exhibited proper development and had no significant anomalies. No fluid was found in the abdomen or thorax cavities. Once a week, the pregnancy was monitored using the same parameters, such as assessing the placental thickening and amniotic fluid levels. The fetuses were checked for mass diameter and HR. After 5 days, the fetus had a mass with dimensions of 0.57 cm along the dorso-ventral axis and 0.22 cm along the craniocaudal axis and with a heart rate of 300 beats per minute (Figure 2 and Figure 3), while the others had a lower heart rate of approximately 220–240 BPM. None of the puppies had free fluid in the abdomen or thorax. 

The pregnancy concluded, and vaginal birth began on day 63 post-ovulation. Five puppies were born; the first two were stillborn, and the others had APGAR scores of 9/10. The litter was homogeneous; the puppies weighed between 170 and 180 g, with no difference between deceased and surviving puppies, and all of them were well-developed. The necroscopy of the two stillborn puppies was performed in the clinic by a pathologist and neonatal expert within 3 h of death, without the newborns being refrigerated. The external and internal examinations of the puppies were within normal parameters for their age and sex. After correct positioning, with the belly facing upwards and the legs splayed, the abdomen was opened: no free fluid was found, and all organs were within normal limits. After that, the thorax was opened; no free fluid was found, and all organs were without macroscopic lesions. Pathologists found a unique difference between the two necroscopy exams in the thoracic cavity. One heart had no macroscopic lesions, and the other had an undeniable bump. The mass was immediately individuated after a simple opening along the sagittal axis. It was a space-occupying mass that expanded from the left atrium to the left ventricle until the distinction between them became impossible. It had a broad base and was pale red, identical to the originating organ (Figure 4 and Figure 5). The right compartments were without macroscopic lesions. 

The heart was preserved in 10% buffered formalin and sent to the laboratory for histological examination. Small sections of formalin-fixed myocardium were paraffin-embedded (FFPE), and 4 micron tissue sections were stained with hematoxylin and eosin (HE), Periodic Acid Schiff (PAS) stain for glycogen, and Masson’s trichrome for connective tissue. Immunohistochemistry was conducted using desmin (Dako, Clone D33) and Alpha-Smooth Muscle Actin (α-SMA-Dako, Clone 1A4) antibodies on a BenchMark ULTRA Ventana system (Medical Systems, Tucson, AZ, USA). PGP 9.5 is a more specific marker for Purkinje cells but was unavailable in the reference laboratory. Immunohistochemical analysis was performed in accordance with the guidelines of the American Association of Veterinary Diagnosticians Subcommittee on Standardization of Immunohistochemistry.

### 2.2. Histopathology, Histochemical, and Immunohistochemical Studies

A poorly demarcated nodule affecting the interventricular septum and the papillary muscles was noted microscopically. It expanded the ventricle walls and protruded into the ventricular cavity, partially reaching the left atrium. The mass had a complex appearance with different populations. The more abundant one consisted of aggregates of large cells surrounded by fibroconnective septa and was frequently distributed around blood vessels. The cells (40–60 µm in diameter) were polygonal and had abundant, pale, eosinophilic to granular cytoplasm (Figure 6). The nuclei were round to oval in shape, located centrally to eccentric, hyperchromatic, and contained small, inconspicuous nucleoli. This population did not display significant anisokaryosis and anisocytosis and no mitoses were noted in 2.37 mm^2^. Intermingled among vacuolized cells, a second population of uniform, spindle-shaped, mesenchymal cells dissecting cardiomyocytes was also present. Within the mass, and especially in subendocardial regions, a multifocal area of necrosis with associated mineralization and macrophagic inflammation with multinucleated macrophages was noted. The cytoplasm of vacuolized cells was multifocally PAS-positive (Figure 7), suggesting a high glycogen content. Connective tissue (highlighted by Masson stain, Figure 8) surrounded most of these cells. IHC revealed a multifocal cytoplasmic positivity to desmin and α-SMA (Figure 9A,B). These histological and immunohistochemical features are consistent with a Purkinje fiber origin. The positivity to α-SMA suggests an embryonic-like phenotype because α-SMA is normally expressed during embryogenesis but disappears after birth [7,8]. 

## 3. Discussion

Cardiac tumors are rare in infants of all species but can occur in swine, cattle, sheep, and dogs [9,10]. In dogs, the percentage of incidence is between 0.12 and 4.33% [3]. In human medicine, the incidence in autopsy studies is approximately 0.01–0.02% [11], with 70% of cases being benign tumors [1]. In infants and young children, the occurrence rate is higher at approximately 0.25% for primary cardiac tumors, whereas autopsy findings reveal a lower incidence (0.002%) [2]. The incidence in live-born infants reported by other authors ranges from 0.02% to 0.08% [12]. Rhabdomyomas are the most common type of cardiac tumors, characterized as benign tumors of striated muscle origin and common in human infants [10]. These are well-described in the human medical literature, while limited information is available for veterinary species [5]. In pigs, rhabdomyomas are frequently reported as an incidental finding and histologically are characterized by so-called “spider cells” that were lacking in the present case [13]. The female bearded seal diagnosed with the tumor in the study by Krafsur et al. was one year old [4]. They also reported that cardiac rhabdomyoma affected a six-week-old fallow deer [14]. The literature on dogs indicated that cardiac tumors primarily affect adult to senior dogs [3,4,5,6,7,8,9,10,11,12,13,14,15], except lymphoma, which is found more often in younger animals [3]. In 2009, two authors described a spontaneous lesion in the heart of a young beagle known as rhabdomyoma. The female dog was 8 months old and included in a routine toxicology survey. The neoplasm showed positive staining for Periodic Acid Schiff in histochemistry, positive staining for desmin and myoglobin, and negative staining for vimentin and smooth muscle actin in immunohistochemistry (IHC) [9]. Cardiac tumors are categorized into two main groups: primary or secondary (metastatic) and benign or malignant. In domestic animals, the most common type of cardiac tumor is hemangiosarcoma, with an incidence of 69%. Many other tumors have been documented to localize in the heart, including aortic body tumors, chemodectoma, paraganglioma, lymphoma, and ectopic thyroid carcinoma. Less frequent heart tumors include thyroid adenoma, melanoma, mast cell tumor, blastoma, granular cell tumor, mesothelioma, myxoma, myxosarcoma, mesenchymoma, undifferentiated sarcoma of presumptive myofibroblastic origin, fibroma, fibrosarcoma, rhabdomyoma, rhabdomyosarcoma, leiomyoma, leiomyosarcoma, chondrosarcoma, osteosarcoma, paraganglioma, peripheral nerve sheath tumor, hamartoma, and lipoma. The canine literature has no consensus regarding the frequency of primary and metastatic cardiac lesions. Ware and Hopper found that only 16% of heart tumors were metastatic, and 85% were primary [3,16]. Aupperle et al. reported that 31% of tumors were primary, and 69% were metastatic [15]. These data are comparable with human medicine clinical records [3]. 

Scientists have recognized different locations of neoplastic lesions inside the heart. In humans, the most common type is bilateral or diffuse neoplastic heart disease. The tumor can affect the left side, right side, and septum or create an intracavitary tumor thrombus [1]. The most common location in infants is the left ventricle, right ventricle, and interventricular septum [2]. Primary heart tumors in dogs and cats mostly originate from the right atrium, with fewer cases found at the cardiac base or in the left ventricle [3]. In our case, the location is between the papillary muscles and the interventricular septum, partially extending into the left atrium. 

The tumor develops in utero and can involve both ventricles and the interventricular septum of the myocardium. In most cases, this disease causes sudden death without giving symptoms before, so the cardiac neoplasm is identified only during autopsy. For this reason, it is fundamental to perform a prenatal ultrasound for early diagnosis, so that the pathological newborn can be effectively handled and surgical treatment for infants can be planned [11]. The typical approach involves palliative treatment with a good prognosis, which can cause regression in size and, in some cases, complete disappearance [11]. Furthermore, in our case, the diagnostic suspicion during pregnancy US was confirmed after birth. In this study, the fetus’s vitality was monitored during the last 20 days of pregnancy. The pathologic fetus exhibited a high cardiac frequency, ranging from 270 to 300 bpm, compared to the other fetuses in the litter. It is not possible to confirm if the affected puppy experienced arrhythmia, but we know that infants can suffer from paroxysmal tachycardias of 280 to 300 beats per minute that lead to death due to cardiac arrest [11,17]. Infants can also exhibit arrhythmias, respiratory distress, congestive cardiac failure, or cyanosis. Cot death is the most common death in babies at one year of life [6]. In many cases, unfortunately, newborns can die without showing any clinical signs. In the veterinary literature, only two case reports contain clinical manifestations caused by cardiac rhabdomyomas: agonal convulsions in a captive fallow deer [14] and chylopericardium and right-sided congestive heart failure in a Staffordshire Bull Terrier [18]. In the reported case of the bearded seal, numerous lesions scattered in all four heart chambers could have caused aberrations in impulse conduction or subclinical arrhythmias [4]. Unfortunately, we could not treat the puppy in our study because, at the time of vaginal birth, it was deceased. To the best of our knowledge, our case is the first case of cardiac tumor diagnosed in a dog fetus during pregnancy. Another diagnostic method is electronic microscopy, which can be useful to highlight the high glycogen content and lack of T-tubules typical of Purkinje cells. It was not performed in our study because the tissue was only fixed in formalin, not glutaraldehyde. 

Cardiac Purkinje cell tumors are extremely rare [6]. This is a tumor located in the conduction tissue inside the heart. The first description dates back to 1965; Voth D Uber defined this lesion as “arachnocytosis of the heart”. The common characteristic of patients is intractable tachyarrhythmias, usually supraventricular in origin, that can lead to death [17]. This type of tumor is documented in the literature as occurring in humans, swine, and bearded seals. It is not clear if a cardiac Purkinje cell tumor is comparable to rhabdomyoma. Jacobsen et al. (2010) proposed the term “purkinjeoma” or “purkinjeomatosis” for cardiac rhabdomyoma in pigs. In their study, two pigs were analyzed, euthanatized, and necropsied. The authors found a nodule, 2 cm in diameter, within the left ventricular wall of the myocardium. The laboratory tests were standard hematoxylin and eosin coloration and immunohistochemistry with antibodies directed against desmin, vimentin, neuron-specific enolase, atrial natriuretic peptide, and protein gene product 9.5 expression (PGP 9.5). The pathologic cells showed cytoplasmic reactivity for desmin and faint reactivity for vimentin, similar to the adjacent normal myocardial and Purkinje fiber cells. This proves that Purkinje cells develop from multipotent myocardial cells that generate myocardial cells during embryogenesis [13]. Perinuclear cytoplasmic granules exhibited reactivity to atrial natriuretic peptide, while reactivity to neuron-specific enolase was more scattered than expected in tumor cells and Purkinje fiber cells. Strong, diffused, and uniform immunoreactivity for PGP 9.5 was found in cytoplasmic rhabdomyoma cells. The study found abnormalities in subendocardial Purkinje fiber cells and myocardial nerve fibers in the affected animals and control pigs [13]. At the end of those studies, the pathologists stated that other investigations confirmed the specificity of PGP 9.5 for adult and fetal Purkinje fiber cells. It is difficult to discern the neoplasm of cardiac myocytes or Purkinje fiber, as well as glycogen storage disease, cellular gigantism, and dysplasia. However, in the authors’ case, the large immunoreactivity for PGP 9.5 encouraged them to propose the term “Purkinjeoma” for the cardiac neoplasm with these immunohistochemical characteristics. Also, the term “purkinjeomatosis” was proposed for the proliferation of other benign tumors (neurofibromatosis or schwannomatosis in human beings, bovine, and horses [13]. In 2014, Krafsur collaborated on a study involving many nodular masses in the heart of a bearded seal; histologically neoplastic cells had PAS-positive material within cytoplasmic vacuoles and cytoplasmic projections. Immunoreactivity for neuron-specific enolase and protein gene product 9.5 (PGP 9.5) led to the term “purkinjeoma” being used to describe this lesion [5]. We performed immunohistochemistry to characterize the lesion. Moreover, the tumor cells exhibited positivity for desmin and smooth muscle actin. We found vacuolized cells often containing PAS-positive material (indicative of high glycogen content found in Purkinje cells) and multiple areas showing positivity to both markers. The lesion in our study could be classified as congenital myocardial neoplasm with complex features and increased Purkinje cells. Notably, in contrast from the case described in a calf [4], where a diffuse increase in essentially normal Purkinje cells was reported, in our case, a focal and complex mass disrupting ventricular architecture is present; thus, the definition of “congenital multifocal increase in Purkinje cells” is not entirely applicable in the present case. Although increased Purkinje aggregates in the myocardium are described as a ‘tumor’, especially in human medicine, the fact that the disease is often reported in infants suggests a developmental anomaly rather than a true neoplasm (like rhabdomyoma and rhabdomyosarcoma). The lack of mitoses and cellular atypia in our case, indeed, support the non-malignant nature of Purkinje cells population. To our knowledge, a complex congenital cardiac tumor with increased perivascular aggregates of Purkinje cells has not been described in dogs, enhancing the importance of this report in the veterinary literature and the need for the accurate surveillance of congenital cardiac masses in this breed. Further experiments, such as electron microscopy and PGP 9.5 immunohistochemistry, are needed to confirm the origin of intralesional vacuolized cell as Purkinje cells with an embryonic-like immunophenotype. 

## 4. Conclusions

To our knowledge, this study is the first documented case in veterinary medicine for the canine species. The Purkinje cells-rich tumor presented in this case shows ultrasound signs and peculiar histopathological characteristics. Additional research and studies are required to clarify the origin of the heart tumor and its classification, particularly in puppies. 

## Figures and Tables

**Figure 1 vetsci-11-00216-f001:**
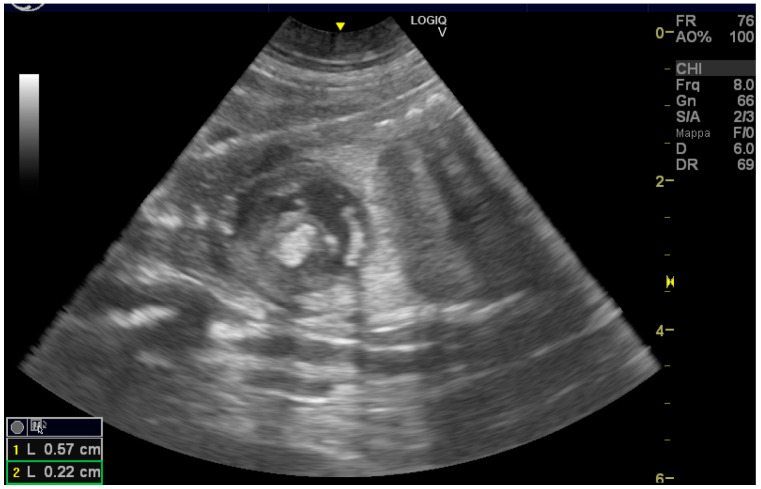
Ultrasound images of the puppy’s heart. (1) Hyperechoic mass occupying the left ventricle.

**Figure 2 vetsci-11-00216-f002:**
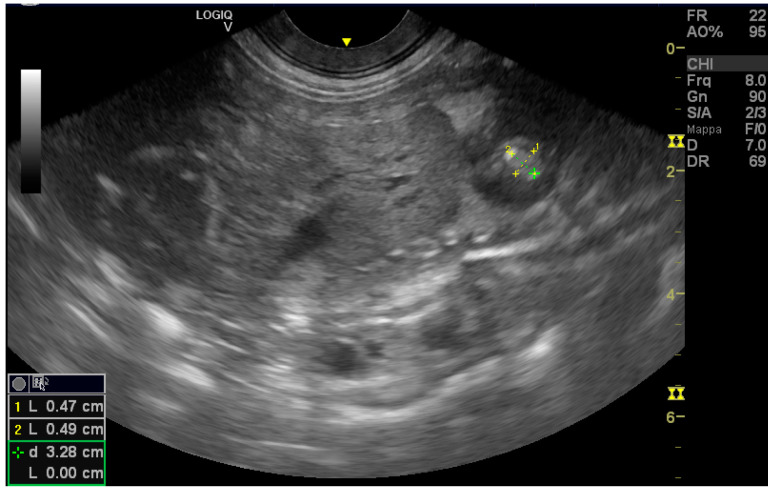
Note the size of the endocardial mass (2) and the elevated heart rate (300 bpm) (3).

**Figure 3 vetsci-11-00216-f003:**
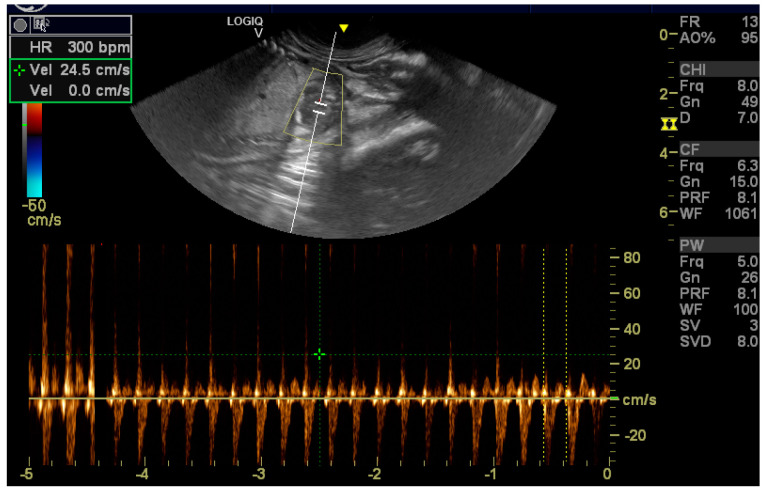
Note the elevated heart rate (300 bpm) of affected puppy.

**Figure 4 vetsci-11-00216-f004:**
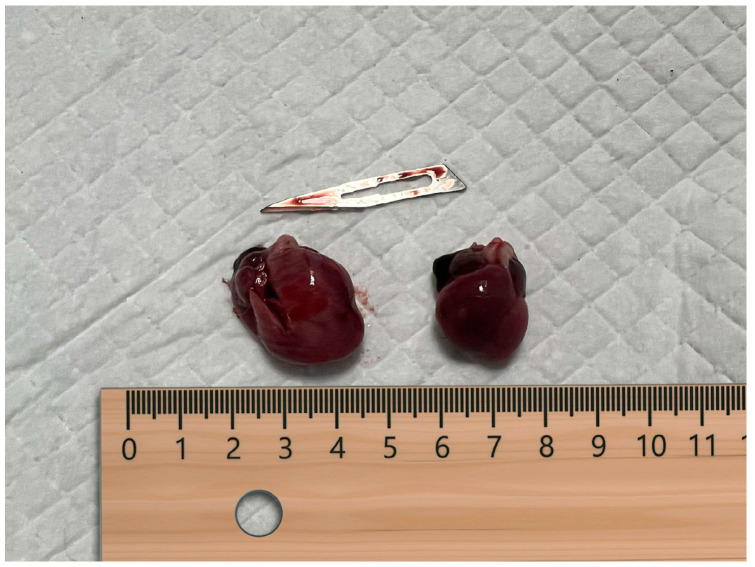
Photo of the hearts of the two stillborn pups.

**Figure 5 vetsci-11-00216-f005:**
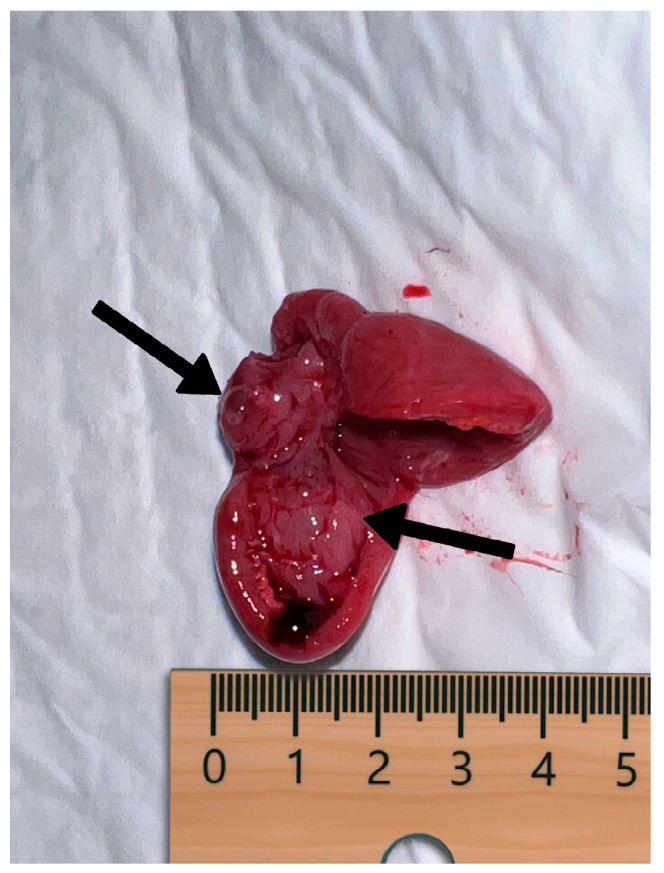
Notice the mass occupying the left ventricle (indicated by arrows).

**Figure 6 vetsci-11-00216-f006:**
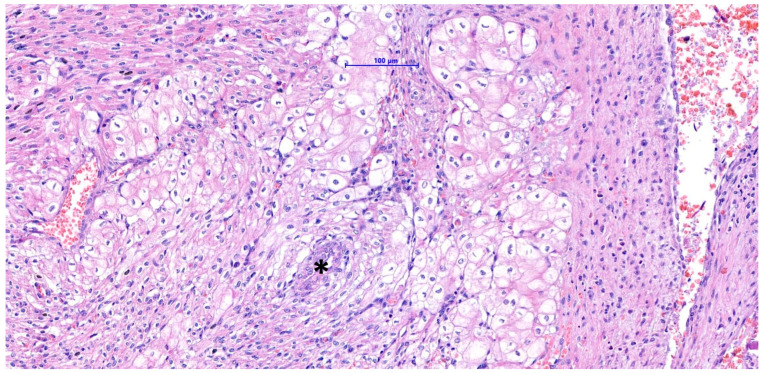
HE (hematoxylin and eosin). The myocardial mass comprises large, polyhedral cells with distinct margins and pale, vacuolated to granular cytoplasm. Note the perivascular arrangement of some cells (asterisk (*) = blood vessel).

**Figure 7 vetsci-11-00216-f007:**
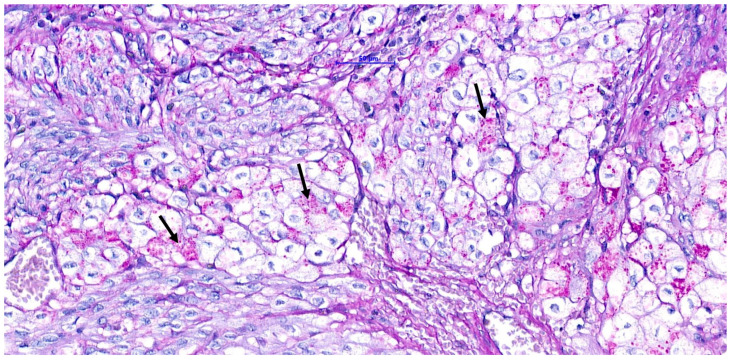
PAS (Periodic Acid Schiff). The neoplastic cells contain PAS-positive granular material (arrows).

**Figure 8 vetsci-11-00216-f008:**
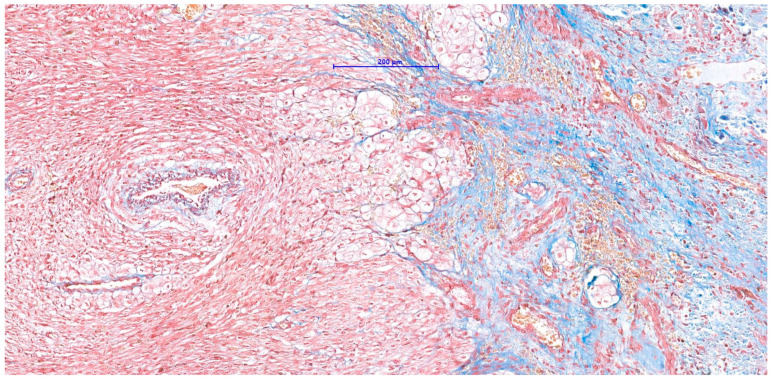
Masson’s trichrome. Connective tissue (stained in blue) surrounds neoplastic cells.

**Figure 9 vetsci-11-00216-f009:**
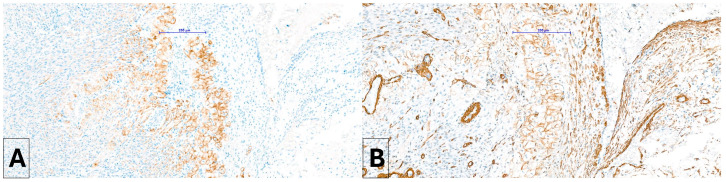
Immunohistochemistry. Neoplastic cells are multifocally positive to desmin (**A**) and AML (**B**).

## Data Availability

All datasets generated for this study are included in the article.

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
