# Peer review of "Ultrasound, Histomorphologic, and Immunohistochemical Analysis of a Cardiac Tumor with Increased Purkinje Cells Detected in a Canine Fetus 42 Days into Pregnancy"

_vetsci, 2024, doi:10.3390/vetsci11050216_

Round 1
Reviewer 1 Report
Comments and Suggestions for Authors
The paper describes a case of Purkinje cell tumour in a newborn dog. Immunohistochemical markers and staining such as PAS and Masson's staining were used to identify the cells. In heart tissue, Purkinje cells and fibers can be well recognized in HE staining, with their typical transparent cytoplasm, their collagen-rich fibers can be seen predominantly in the Masson Goldner Trichrome, being a low cost alternative to collagen I immunohistochemistry (Bürgisser Acta Histochem 2023 Jan;125(1):151993). The report is interesting and worth publishing. The antibodies used by the authors in their work do not give the certainty of differentiating cell origin. But immunohistochemistry combined with morphology, PAS and Masson's staining allow reliable identification. It would have been useful to use electron microscopy to identify the cellular origin of the neoplasm. I think it is useful that the authors mention this possibility. Histological images are important in this work. The ones included are not of good quality, in some cases they appear out of focus. The macro image of the lesion is of poor quality, consider improving it by inserting a ruler or something else as a metric reference instead of the scalpel blade. Also, it would be useful to include image of Masson's stain. The staining methods must be described in more detail, in particular for immunohistochemistry dilutions and chromogen used
Author Response
The authors are grateful for the suggestions
- We Mention this possibility to use Electronic microscopy in 176-178 lines
- We change Micro and macro image
Reviewer 2 Report
Comments and Suggestions for Authors
The paper describes a heart nodule (Purkinje cell tumor) in a newborn puppy dog, using histology and routine immunohistochemistry. Despite the authors' efforts and the enthusiasm that shines through, the case study did not have the scientific rigor to confirm the diagnosis. Moreover, the text requires a thorough linguistic and structural revision.
Title:
The title leaves no space for other differential diagnoses. I prefer a title like “heart mass or area in the fetus”. Using the terms "First record" or "never described" in a title implies a complete definition of the diagnosis in light of exhaustive literature on the subject. It does not seem that any of these conditions are met.
Simple summary. the simple summary is longer and more complex than the summary.
Key words. Remove Numbers
Introduction. The introduction cannot be made only on tumors of the heart and Purkinje cell tumor. The introduction should be more general and introduce other hypotheses. Eventually, the discussion can be focused on the tumor.
Case description. The text is fluent and narrative, lacking measurements and related statistics. For example, the increase in the heart rate of the fetus affected by the malformation is interesting, but what was the range, how many measurements were made, were the differences significant when compared to those detected in other fetuses, and what test was used? The necropsy description is also a bit sparse. It seems that the cadaver has been sent frozen or refrigerated to a diagnostic server, please indicate it. The histological description should be improved.
Citations and comments should be avoided in the clinical case, which instead must find the right space in discussions. The ultrasound table is too small, perhaps it is preferable to reduce the images to 1 or 2 but put them larger. The same goes for the macroscopic picture, which is too small and not legible (use arrows, and circles to visualize the mass). Histological pictures have a low resolution, please show some diagnostic details with a HE stain.
Discussion. The authors are sure that the mass is a tumor, even though it is in a fetus, in an unusual location, an entity never described in dogs. Jacobsen's work reports intramural multifocal nodules and cells very different from those in the reported case and concludes that an antibody panel including PGP9.5 is needed to "propose" the diagnosis of Purkinje cell tumor. Moreover, the authors do not include SMA among the antigens. SMA is not a marker of Purkinje cells but may have many other origins in the newborn’s heart.
Comments on the Quality of English LanguageExtensive editing is required.
Author Response
The authors are grateful for the suggestions
- LINES 114-116 changed as:
In Human medicine the incidence in autopsy studies is about 0,01% [6]- 0.02 %, 70% of these are benign tumors [1].
- We add the paper about bearded seal in these lines:
167: Modified as suggested
208-212: Modified as suggested
- We improved quality of images, we hope now is good quality
- We changed the caption of figure 2 (now Fig,5)
Photos of the hearts of the two stillborn pups. Notice in the left one, the mass occupying the left ventricle
- We delayed lines 207-208
Reviewer 3 Report
Comments and Suggestions for Authors
The present case report of a Purkinje cell heart tumor in a puppy is very interesting, both due absence of previous reports of this type of tumor in this species, and to its detection in utero.
However, the article needs extensive English revision by a native, for clarity and scientific rigor. In the present form, it is difficult to read and understand fully, and therefore, only after extensive revision, including summary, abstract, introduction, case description, and conclusions, a proper review will be possible. The discussion is confusing, mainly due to language and its structure could be improved by aiming for clarity. For example, in lines 114 to 116 it is said that "In Human medicine the incidence in autopsy studies is about 0,01% [6]- 0.02 % [1]. Among this group, 70% of these are surgically excised and are benign tumors [1]." It is not immediately clear how 70% of tumors are surgically excised in autopsy studies.
The authors may choose to also cite the following paper, since it refers to a case report in a mammal, with a good description and discussion of histopathology findings.
Krafsur, G., Ehrhart, E. J., Ramos-Vara, J., Mason, G., Sarren, F., Adams, B., ... & Duncan, C. (2014). Histomorphologic and immunohistochemical characterization of a Cardiac Purkinjeoma in a Bearded Seal (Erignathus barbatus). Case Reports in Veterinary Medicine, 2014.
The quality of the images must be improved, especially concerning the macroscopic and optical microscopy images (Figures 2 and 3).
In Figure 1 the hyperechoic mass is described as being located in the left ventricle, while in Figure 2, it is mentioned as being in the right ventricle. This should be clarified.
Lines 207 and 208 are probably typos.
Comments on the Quality of English Language
he present case report of a Purkinje cell heart tumor in a puppy is very interesting, both due absence of previous reports of this type of tumor in this species, and to its detection in utero.
However, the article needs extensive English revision by a native, for clarity and scientific rigor. In the present form, it is difficult to read and understand fully, and therefore, only after extensive revision, including summary, abstract, introduction, case description, and conclusions, a proper review will be possible. The discussion is confusing, mainly due to language and its structure could be improved by aiming for clarity. For example, in lines 114 to 116 it is said that "In Human medicine the incidence in autopsy studies is about 0,01% [6]- 0.02 % [1]. Among this group, 70% of these are surgically excised and are benign tumors [1]." It is not immediately clear how 70% of tumors are surgically excised in autopsy studies.
The authors may choose to also cite the following paper, since it refers to a case report in a mammal, with a good description and discussion of histopathology findings.
Krafsur, G., Ehrhart, E. J., Ramos-Vara, J., Mason, G., Sarren, F., Adams, B., ... & Duncan, C. (2014). Histomorphologic and immunohistochemical characterization of a Cardiac Purkinjeoma in a Bearded Seal (Erignathus barbatus). Case Reports in Veterinary Medicine, 2014.
The quality of the images must be improved, especially concerning the macroscopic and optical microscopy images (Figures 2 and 3).
In Figure 1 the hyperechoic mass is described as being located in the left ventricle, while in Figure 2, it is mentioned as being in the right ventricle. This should be clarified.
Lines 207 and 208 are probably typos.
Author Response
The authors are grateful for the suggestions
Title: Ultrasound, Histomorphologic and Immunohistochemical description of a Cardiac Purkinjeoma detected in a foetus of 42 days of pregnancy
Simple summary: The summary has been modified and simplified as suggested
- Introduction: The introduction has been modified as suggested
- Add in the text: cardiac range, how many measurements we made, the difference between ill and safe puppy;
Line 65: The necroscopy was done in the clinic, by pathologist and neonatal expert. No one puppies was carried out or freezed.
necroscopy: modified and improved as suggested
Line 77 we add: Right compartments were without macroscopic lesion.
- Delete citation and comment in clinical case: Done
Line 46 we have replaced “parameters were normal “with “within physiologic limits”
Line 48 we delated “within normal range” and wrote “Progesterone concentration was made by Tosoh AIA360 (Japan), due to a history of hypoluteinism. The results were 34 ng/ml at time of diagnosis and 36 ng/ml one week later.
Line 49 we delated “parameters within normal range” and wrote “The US at 42 days of pregnancy detected these parameters…”
Line 51 we delated “strange” and wrote “abnormal”
Line 51-55 modified
Line 57-60, modified.
Line 62 we delated “Pregnancy was going well” and Now we wrote: Pregnancy was concluded and..
Round 2
Reviewer 1 Report
Comments and Suggestions for Authors
The article has been improved and the pictures are of reasonable/good quality.
Author Response
Thank you very much for the valuable advice you had givenReviewer 2 Report
Comments and Suggestions for Authors
The text has improved considerably. However, structural problems still exist. The introduction is short, while the discussions do not discuss the clinical case but give information on heart tumors. Sentences that could be moved to the introduction. In the introduction, there are sentences related to the clinical case and this is not acceptable, except the final sentence (aim). The clinical case is well presented. Small corrections are required (heart rate is reported with different values, 3 times). In the clinical case, there must be no discussions (PGP 9.5). The discussions are excessively long (improve the introduction). The fact that the authors do not do electron microcopy and PGP 9.5, as they denounce, does not allow them to diagnose a Purkinje cell tumor. If the article had the title "cardiac mass" or even "cardiac tumor", it might be accepted (although in my opinion, it is not a tumor). In discussions, the authors could hypothesize the presence of a Purkinje cell tumor.
However, if the authors are sure of the diagnosis, further experiments are required and then the article must be rejected, according to the MDPI guidelines.
Comments on the Quality of English LanguageMinor revisions are required
Author Response
Thanks for your suggestions. In the discussion we provided a more extensivereasoning about the non malignant nature of the mass, suggesting a developmental anomaly rather than a true neoplasm.
Reviewer 3 Report
Comments and Suggestions for Authors
The authors report a very interesting and rare case of a primary cardiac tumour in dog, rich in Purkinje cells, detected in utero during ultrasound examination and the present document presents an improved version. Introduction and discussion are more complete and references are adequate.
Still, there are language and clarity issues that affect the quality and clarity of the summary and the clinical case description. Before being accepted for publishing I think proofreading by a native English is necessary. Some examples:
Line15 and 16: "cels" instead of "cells" as in "with increased Purkinje cels in a mainly perivascular fashion"; do the authors mean Purkinje cells were increased in number and concentrated in perivascular areas?
Line 17 "cardiactumor" instead of "cardiac tumor"
Line 28 and 43: "The tumor is found (..) but never in" it would be more correct to say"This type of tumor has been described in"
Lines 56 -61 "On day 42 of pregnancy, the US detected the following parameters: a fetus with a placenta thickness of less than 7 mm, anechoic liquid in all neonates without echoic smoke, and all fetuses showing growth rates consistent with the pregnancy date. Only one of the five fetuses presented an abnormal, echoic mass inside their heart (Fig.1). It was located in the left atrioventricular junction, displaying a soft fringe and a light shadow cone. The diameter measured was 0,49 cm on the puppy’s dorso-ventral axis and 0,47 on the craniocaudal axis" In this paragraph it is not clear if the foetus ("a fetus with a placenta thickness of less than 7 mm") was the same one with the mass.
Line 59 “Only one of the five fetuses presented an abnormal, echoic mass inside their heart” should be corrected to “Only one of the five foetuses presented an abnormal, echoic mass inside its heart”.
It is not mentioned when the measurements in lines 67 and 68 were done if the text in lines 67 and 68 "One fetus had a mass with dimensions of 0,57 cm along the dorso-ventral axis and 0,22 cm along the craniocaudal axis", since they differ from measurements referred in lines 61 and 62.
I do not understand what the authors mean by “Neither of the puppies were removed or frozen.”. Perhaps this refers to the necroscopy being done in the same clinic where the birth took place, but it would probably be better to refer how long after the birth the necroscopy was done and whether the cadavers were refrigerated.
The authors in line 82 write “No maternal aggression was observed in either case.” while previously stating the first two puppies were stillborn. Do the authors wish to further reinforce that the puppies did not die due to maternal aggression?
Line 185 “being identified only during autopsy cardiac.” Consider revision and correction.
Line 189 “Further, in our case, we can confirm the diagnostic suspicion during pregnancy using US.” Consider rephrasing. Do the authors mean they were able to confirm postmortem the diagnostic suspicion of a cardiac tumour established during the US done during pregnancy?
Line 191 “Pathologic fetuses exhibited a high cardiac frequency, ranging from 270 to 300 bpm, compared to the other fetuses in 192 the litter”- How many foetuses presented high cardiac frequency? One foetus? Two foetuses?
Line 193 “We do not know if a puppy experienced” considerer correcting to “It is not possible to confirm if the affected puppy experienced “
Line 263 consider correcting "richtumor" to "Purkinje cells-rich tumor"
Regarding images: Scale bar in figure 9A is not readable; scale bar in figure 8 is barely legible due to colour.
Comments on the Quality of English Language
As above mentioned, proofreading by an English native is strongly recomended.
Author Response
The authors thank you for your attention and further clarifications.
Unfortunately the images cannot be edited
Line 15-16: Correct
Line 17: correct
Line 28 and 43: correct
Lines 56 -61: "a fetus with a placenta thickness of less than 7 mm" was a typo from the previous version: placental thickness was normal (<7mm) for all fetuses. Thanks for noticing
Lines 56 -61 : We specified that the second measurement is 5 days later
Line 59: correct
Line 62: explained better
Line 82: Yes just like this, but since it's not important we'ldelete it
Line 185: Revised
Line 189: explained better
Line 191: only the foetus whith the mass presented high cardiac frequency
Line 191: changed
Line 263: Correct
For English, we actually paid for the magazine review that was done
Round 3
Reviewer 2 Report
Comments and Suggestions for Authors
The simple summary has been modified but some small spacing and spelling errors (cels) remain. The first line does not agree with the title.
The summary presents the case report. The diagnosis does not agree with the title. Some spacing errors are present. There is no introduction and poor discussion.
The introduction is sparse and confusing. it starts with the human species, then the dog, then humans again, and then the animals. There are discussions about the clinical case, but the reader does not know the clinical case. The aim does not match the title.
The clinical case is well described. However, the text and images do not agree. In the test, at least two checks were described, but the dates of the ultrasound pictures do not follow this chronology. The second is from a month earlier in another year, while the first and third are on the same day, in the text they should differ by a week. In "clinical findings", there are methods and descriptions of histopathology that should be moved. There must be no discussion in the results (such as the PGP comment). The reference to the guidelines is out of place, among other things without citation. Citations 6 and 7 are not allowed in the results, as they do not refer to methodologies but are comments.
The discussions are very long. Some generic sentences about tumors could find space in the introduction. No differential diagnoses are taken into consideration.
The references have severe problems with style.
Comments on the Quality of English LanguageEnglish has improved significantly.
Author Response
The authors thank you for your attention and further clarifications.
We have corrected the diagnosis in both the Summary and the Abstract and also in the Aim.
We have eliminated the comment on the clinical case from the introduction.
The inconsistency in ultrasound dates is caused by the fact that because it was pointed out to us that a photo was of low quality we took a clearer frame from one of the ultrasound videos.
However, the ultrasound machine saved the date of the frame obtained.
Thanks for noticing. We have removed the dates from the images to avoid confusion for those who will read the paper.
The request to shorten the introduction and broaden the discussion, as well as the clarification to move some sentences, came to us from another reviewer during the first revision.
Since 2 reviewers accepted the paper as is, we cannot now change it again.